# National Implementation of Perinatal Mental Health Treatment—The MumSpace Digital Stepped-Care Model

**DOI:** 10.3390/ijerph22030361

**Published:** 2025-02-28

**Authors:** Jeannette Milgrom, Brydie Garner, Andre Rodrigues, Jane Fisher, Julie Borninkhof, David Kavanagh, Alan W. Gemmill

**Affiliations:** 1Parent-Infant Research Institute, Austin Health, Heidelberg, VIC 3084, Australia; brydie.garner@austin.org.au (B.G.); andre.rodrigues@bupa.com.au (A.R.); alan.gemmill@austin.org.au (A.W.G.); 2Melbourne School of Psychological Sciences, University of Melbourne, Melbourne, VIC 3010, Australia; 3School of Public Health and Preventive Medicine, Faculty of Medicine, Nursing and Health Sciences, Monash University, Melbourne, VIC 3800, Australia; jane.fisher@monash.edu; 4PANDA Perinatal Anxiety & Depression Australia, Fitzroy, VIC 3065, Australia; julie.borninkhof@panda.org.au; 5Centre for Children’s Health Research, Queensland University of Technology, South Brisbane, QLD 4101, Australia; david.kavanagh@qut.edu.au; 6School of Psychology and Counselling, Queensland University of Technology, Kelvin Grove, QLD 4059, Australia

**Keywords:** depression, perinatal, postnatal, digital, treatment, implementation

## Abstract

Perinatal depression is highly prevalent, yet there is a very low rate of treatment uptake and help-seeking. The *MumSpace* Initiative was funded by the Australian government to invest in digital stepped-care treatments and support for perinatal depression, to improve mental health outcomes and national access. This paper describes the reach of the *MumSpace* initiative as a one-stop shop offering perinatal depression treatments with a solid evidence base (MumMoodBooster programmes), supported by a prevention programme addressing modifiable risk factors through a smartphone application (*MindMum*) as well as evidence-based universal prevention programmes. We have brought together multi-skilled teams and a Perinatal Depression Consortium to deliver the programmes and address changing technology. The effectiveness of *MumSpace* was evaluated through systematic monitoring of consumer reach: data analysis of website traffic and resource uptake. *MumSpace* has successfully sustained engagement, attracting over 275,000 visits since its launch in 2017, with the number of visitors to the website increasing year on year. The central treatment tools, *MumMoodBooster* and *Mum2BMoodBooster*, have reached over 10,000 Australian women, largely through self-referral. Despite the development of a portal for direct clinician referral and monitoring, continuing challenges for implementation involve integrating digital treatments into traditional services and recruiting professionals to directly engage mothers.

## 1. Introduction

### 1.1. Human and Economic Cost: The Burden of Perinatal Depression and Anxiety

The human and economic cost of perinatal depression is substantial.

An estimated 20% of women experience depression either antenatally or in the first postnatal year (perinatal depression) [1]. Prevalence estimates vary depending on the instrument used to measure depression or whether point or period prevalence is reported, but in general this average prevalence rate has been reported in repeated studies with higher figures for self-reported measures, disadvantaged groups, and lower-income countries [2,3,4].

Maternal depression has profound negative consequences on a woman’s well-being, impacts her partner and their relationship [5,6], and is often accompanied by significant anxiety. In addition, children followed through infancy, adolescence, and adulthood have been shown in a number of studies, including our own, to have a substantially higher risk of enduring mental health, cognitive, and behavioural problems [7,8,9,10].

Perinatal mental health problems also damage economies. A definitive analysis by the London School of Economics has demonstrated that perinatal mental health problems together cost GBP 8 billion for every 1-year cohort of births in the UK. Importantly, nearly three quarters of these costs are due to the enduring negative impact on children into adulthood rather than the impact on mothers themselves [11]. In Australia, analyses of estimated costs per 1-year cohort are similar [12].

### 1.2. The Service Gap and Need for Accessible Treatment

The current health system in many countries is inadequate to meet the need and demand for perinatal mental health services. In Australia [13] and other counties [14], the system’s response has often resulted in a patchwork of services [15] with professionals often unable to identify appropriate referral pathways [16].

Currently, help-seeking and treatment rates are unacceptably poor, and even when services engage in screening, identification is not well-integrated with subsequent management [17]. Only 1 out of every 10 women are adequately treated to remission even when identified [18,19,20].

In addition to a lack of available and affordable services, barriers to treatment include ongoing stigma associated with perinatal mental health problems, which may result in women avoiding seeking diagnosis, treatment, and care [21]. Practical obstacles include difficulty getting to clinic appointments due to the time demands of managing an infant, compounded by symptoms of depression such as lethargy and the inability to concentrate [22].

### 1.3. Advantages of Digital Service Delivery

Access to digital tools has the potential to reach women at a population level, bring greater consistency and coherence to services in perinatal mental health, and improve uptake. Evidence-based digital tools can be provided as standalone treatments, as part of hybrid treatment with traditional services, and with or without guided support.

The value of e-solutions for this vulnerable population includes 24/7 access, provision of cost-effective, affordable services, and the ability to deliver tailored, scalable support regardless of location. Additionally, digital tools may reduce stigma associated with seeking support.

### 1.4. Steps to Population-Based Implementation of Digital Treatment for Perinatal Depression

#### 1.4.1. Developing an Evidence Base for E-Treatment of Perinatal Depression

In order to credibly treat depression at a population level with digital tools, we need to have well developed and rigorously evaluated e-treatments. Until relatively recently there were (to our knowledge) no evaluated perinatal depression-specific online cognitive–behavioural treatment programmes for women with a diagnosed depressive disorder. In collaboration with colleagues at the Oregon Research Institute, we engaged in a 10-year programme of development and consumer testing, a feasibility trial and multiple randomised controlled trials with National Institutes of Health funding and National Health & Medical Research Council support [23,24,25]. We based the programme on our *Getting Ahead of Postnatal Depression* (GAPND) programme, a face-to-face group treatment developed with consumer input incorporating key components of the Lewinsohn’s Depression Course, which was successfully trialled in numerous studies [26]. We used a systematic, iterative development process following the Science Panel on Interactive Communications and Health guidelines [27] consistent with a staged approach for the development and testing of behavioural interventions [28,29]. Developmental work began with focus groups of women from the target population and direct user testing of prototype elements. In the web-based programme, *MumMoodBooster*, the core CBT content of GAPND is delivered in six weekly sessions and is supplemented by a partner website and library articles. Sessions mimic the process of face-to-face therapy and include personalised interactive exercises, videos, and feedback on progress. Following user testing and a feasibility trial [23], the first randomised controlled trial (RCT) of *MumMoodBooster* [24] involved women with DSM-IV diagnoses of major or minor depression.

*MumMoodBooster* reduced depressive symptoms with a large effect size (d = 0.83). Diagnostic interviews at the end of treatment demonstrated a four-fold increase in remission for women allocated to the *MumMoodBooster* digital depression treatment programme (80% recovery rate compared to controls).

*MumMoodBooster* was then evaluated directly against face-to-face CBT (using the GAPND content in an individualised format) and treatment as usual in a 3-arm RCT [25].

Previously published trials mostly used “waitlist” or “routine care” comparators only. This three-group parallel RCT established that treatment with *MumMoodBooster* was at least as good as face-to-face CBT in producing remission and was statistically superior in the speed of treatment response and reduction in symptom severity. It marked an important breakthrough in our knowledge and for establishing the clinical validity of e-treatments for serious perinatal depression and increasing accessibility in real-world applications.

Following this, Gemmill and colleagues [30] reported a successful feasibility study of an antenatal version, *Mum2BMoodBooster*, in a cohort of 27 depressed pregnant women, of whom 21 had current diagnoses of major or minor depression on the DSM-IV. Again, significant reductions in depressive symptoms were observed, with large effect sizes [30]. In addition, two other US implementation and RCT studies of a US version (*MomMoodBooster*) have yielded similar results and effect sizes [31,32].

#### 1.4.2. Developing a Model of Care

Having developed effective treatments for perinatal depression, we responded to an Australian federal government tender calling for the provision of a digital, self-help perinatal depression treatment programme for women suffering from perinatal depression and a smartphone application (app) for ‘at-risk’ perinatal women, to reduce and ameliorate the impact of depression.

In response, we developed a model of care embedding the treatment programmes in a central platform that would serve as a one-stop shop for women who self-identify as depressed, as well as those who might benefit from preventive or universal tools.

We aimed to provide digital options allowing women to self-assess their level of need, and for those who needed immediate help for significant depressive symptoms, easy access to specialised treatment programmes. By also providing tools for prevention and mental health promotion that apply to the whole population, the delivery of *MumMoodBooster* occurred within a digital stepped-care approach able to offer tools for differing levels of need and the ability to step up or down (see Figure 1). All tools had links to helplines and encouragement to seek professional assistance when needed, with guided support offered to women at higher risk and engaging in the treatment programme.

We brought together a Perinatal Depression e-Consortium (PDeC) consisting of academics, clinician researchers, service providers, and consumer organisations with established records of developing and providing evidence-based digital resources and support for perinatal mental health. We were successful at gaining government funding, which allowed these resources to be freely available to all Australian women.

Our model of care offered the following core components:

##### Treatment

Our web-based tool, *MumMoodBooster*, was market-ready, effective, and rigorously developed. As described, its components had been evaluated in numerous trials, including useability, feasibility, and randomised controlled trials with a solid evidence base demonstrating its effectiveness. It provided a targeted, digital, self-help perinatal depression treatment programme for women with a diagnosis of perinatal depression. In addition, we offered an optional coached component to *MumMoodBooster* for women at higher risk (EPDS greater than 15). Coaching resources were delivered by PDeC partner JB. In this way, at its highest level, the stepped-care approach invited women to step up to coach-assisted online support by low-intensity workers with the aim of providing a safety net and increasing engagement and motivation. For all women, motivational SMSs were provided and risk management alerts were in place to ensure safety, based on assessment questions. Women were guided towards an appropriate referral, including the provision of 24/7 crisis numbers. Finally, antenatal women could opt for the pregnancy-specific treatment, *Mum2BMoodBooster.*

##### Prevention

A new smartphone app (*MindMum*) was developed by PDeC partners, suitable for women who self-identify as having adjustment difficulties. The intervention is considered an indicated preventive measure, designed for women who have subsyndromal symptoms of depression and anxiety or have risk factors which make them vulnerable to depression. The app provides assistance which may prevent further escalation, as well as the ability to ‘step up’ to the treatment programme for early intervention. Women are able to monitor their emotions, choose self-paced guided activities designed to create behaviour change, and engage in positive thinking. In this way, women are assisted to identify their symptoms early and develop coping strategies. It combines tip sheets, ideas on how to improve mood, and audio on mindfulness and relaxation. Users can create a personalised plan for action, monitor mood and record notes, obtain feedback on their progress, and share their work with others. The app was developed prior to the implementation of the stepped-care model. Revised versions have since then been developed based on lived experience feedback and ongoing evaluation (to be reported in a separate paper).

##### Universal Programmes

Within the stepped-care model, we also have available programmes suitable for all parents and aimed at a population-wide application to address common problems affecting all new parents, such as sleep and settling, common stressors in the first year of life that may contribute to a depressive episode. These universal programmes also address risk factors for depression and therefore can be also considered preventive. Universal programmes apply to a larger proportion of the population, and therefore, even small improvements can have a significant impact.

*What Were We Thinking* (WWWT) was developed by a research group led by our consortium partner (JF). WWWT [33] is an interactive website addressing two potentially modifiable risk factors for postnatal mental health problems: non-adversarial renegotiation of roles and responsibilities between intimate partners and skills to manage unsettled infant behaviours. The programme is unique in being explicitly inclusive of women and men caring for their first baby. The in-person psychoeducational programme integrated into first-time parents’ groups proved effective in reducing postnatal depression and anxiety among primiparous women at 6 months and was sustained at 18 months. WWWT has been evaluated in RCTs [33,34].

The *BabySteps* web programme, which was developed by a research group led by another of our consortium partners (DK), is a universal self-guided tool that supports the positive perinatal adjustment of both mothers and fathers. The programme provides tips on baby care issues, self-care, and maintaining positive and mutually supportive relationships and includes a segment especially for fathers. Each specific tip can be used to create a stepwise plan on its implementation. Users are also encouraged to add photos of themselves and their baby to remind them of positive experiences, and the landing page features a different quiz question and tip each time they open it. *BabySteps* has been successfully trialled in an RCT [35] and evaluated as having a promising cost-effectiveness profile [36].

##### Psychoeducational Resources

We provided psychoeducational tools about the symptoms of perinatal depression and what each programme offered to allow for self-assessment and multiple points of entry for the user journey. Each digital tool included a ‘get help’ or ‘more information’ button to enable users to progress through the options.

### 1.5. Aims of This Paper

Through this paper, we aim to provide a description and evaluation of the national reach of the *MumSpace* initiative’s central website and the resources it houses and to describe patterns of user and stakeholder engagement, promotional activities, as well as quality improvement initiatives.

The target population was conceived as deliberately broad, aiming to include all women of child-bearing age in Australia who have a young child or who are expecting a child. Resources were designed to be suitable to those with no current mental health difficulties, mild difficulties, or more elevated depressive symptoms.

## 2. Materials and Methods

### 2.1. Project Implementation Framework

The Project Lead reports regularly to the Department of Health, the funding agency, and ensures six-monthly reporting against the key performance indicators of reach, stakeholder involvement, risk management, and promotion. Implementation is supported by an Implementation Research Manager who coordinates teams of support staff for each resource under the oversight of a Project Lead. Teams include programmers and a marketing manager, supported by the collective advisory input of the Perinatal Depression e-Consortium (PDeC). Regular project and governance meetings (weekly reporting, discussions, and decision making) between the Project Lead, Managers, and operational staff ensure close monitoring of performance and early identification of problems and barriers to ongoing implementation as well as quality improvement opportunities. Led by the Parent-Infant Research Institute (Project Lead JM) as the administering institution, the PDeC partnership is governed by multi-institution agreements and includes Global and Women’s Health, Public Health and Preventive Medicine, Monash University (JF), the ePsych research group, Queensland University of Technology (DK), and Perinatal Anxiety and Depression Australia (JB) and others previously associated with these organisations (see Acknowledgements), as well as Jean Hailes for Women’s Health. Monthly meetings of this group provide advice on the project and responses to strategic issues by the collective, ensuring that the core values and principles guiding service delivery are adhered to, including commitment to evidence-based practices, inclusivity, accessibility, involvement of consumers, and ethical considerations. Additional publication-focussed or subcommittee meetings are held as needed.

### 2.2. Quality Improvement Cycle and New Developments in This Study Period

By providing regular monitoring of website and resource performance, this structure facilitates continuous quality improvement. For example, during the period reported here, based on consumer feedback, various updates to the *MindMum* app have been released on the App Store and Google Play, and work has resulted in developing a dedicated clinician’s portal for *MumMoodBooster* and *Mum2BMoodBooster* and in the commencement of construction of a more mobile-phone-responsive version of the *MumMoodBooster* programme.

A clinician portal was developed to allow for direct referrals from health professionals with the capacity to monitor women’s progress through the programme, in addition to the option for women to join directly. General Practitioners (GPs) and other professionals can register and receive a code to give a woman as a referral and, with her consent, receive an extended mental health assessment upon account registration and risk alerts if a woman’s depression scores escalate or she is experiencing thoughts of self-harm. Women’s progress through the programme can also be monitored so that this treatment can form part of the clinician’s treatment plan, offering a hybrid model of intervention.

### 2.3. MumSpace Website and Architecture

A central website, *MumSpace* (mumspace.com.au: accessed on 19 February 2025) is maintained to act as a vehicle to house and disseminate the central resources needed to deliver the aims of the overall initiative. Programmes are accessed through this one-stop website allowing women to identify their own emotional health needs and choose the level of care that best suits them (see Figure 2).

The organising principle for the structure of the central website, *MumSpace*, is the stepped-care model described above and includes resources suitable for all women who are pregnant or have recently given birth. Our specialised interventions and treatments for those with more severe difficulties and mental health problems are supported by less intensive resources and by universal support relevant to all new mothers. All resources were evidence-based, had consumer co-design in their development stages, and were tested for comprehensibility and acceptability. In addition, these resources achieved HealthDirect and Health on The Net accreditation and are listed on the federal government’s digital mental health site, https://www.medicarementalhealth.gov.au/ (accessed on 19 February 2025).

The stepped-care structure catering for different levels of need is mirrored in the architecture of the *MumSpace* website itself. Four main pages follow the logic of the model and provide access to support at the appropriate level and the ability to move between steps in the model. Users can enter at any step and move up or down. A ‘Self-Assessment’ page provides information and appropriate links to help users locate the resources that suit their needs best.

The ‘For All New Parents’ page provides information and access to two universal programmes (*What Were We Thinking* and *BabySteps*) designed for use by couples with young infants or by couples expecting a child.

‘When You Need Extra Help’ is intended for perinatal women who are experiencing some difficulties or by those who are interested or concerned about their mental well-being in the perinatal period. Information and access links to the *MindMum* smartphone app is housed on this page.

Finally, representing the highest level of e-intervention for the stepped-care model is the page entitled ‘Online Treatments’. On this page, information is presented about what potential registrants can expect from the two digital cognitive–behavioural therapy programmes, *MumMoodBooster* and *Mum2BMoodBooster*, and the credentials of their scientific evidence base. Links to the registration pages of these two treatment programmes provide direct access to interested users. Once registered, users who score high on the EPDS are offered online support through contact from a telephone coach.

In addition to these four central pages, mumspace.com.au also includes a ‘Home’ page, a ‘Resources’ page including downloadable tip sheets and brochures, an ‘About’ page, and an ‘Urgent Help’ page containing links to immediate emergency resources. On the Home page, new information and features were added during implementation, including links to the clinician portal.

Additionally, *MumSpace* has social media accounts on Facebook and Instagram to help increase awareness and facilitate dissemination to consumers, health agencies, health professionals, and other stakeholders.

### 2.4. Sources of Data

A major purpose of this study was to quantify the national implementation, reach, and maintenance of the *MumSpace* resources across Australia from launch in 2017 until 2024. This focus was consistent with 3 of the RE-AIM Metrics (Reach, Implementation, and Maintenance; [37]) providing a useful and widely used conceptual rubric for measuring the impact of health system innovation. We report on dissemination and uptake by health service consumers (Reach), efficiency of integration into practice (Implementation), and the sustainability of improvements to enable ongoing monitoring and quality improvement (Maintenance).

To measure this, Google Analytics was used to collect data on *MumSpace* website usage. Information about social media engagement was extracted from Facebook and Instagram accounts using the functionalities available on those platforms. Registration data were exported from the administrative dashboards of *MumMoodBooster*, *Mum2BMoodBooster*, *MindMum*, *What Were We Thinking*, and *BabySteps*.

### 2.5. Data Extraction

We used Google Analytics to track activity on *MumSpace* from its formal launch in October 2017. For the present study, we extracted data from October 2017 to 30 June 2024.

Specifically, for the *MumSpace* website, we extracted data on the number of sessions (visits), number of page views, average pages viewed per session, average duration of website visits, most frequently visited pages, device type used to access website, geographic location (state/territory within Australia), number of click-throughs from *MumSpace* to treatment resources, and user acquisition (e.g., organic search, referral from external website, etc.).

For *MumMoodBooster* and *Mum2BMoodBooster*, we extracted data on total registrations, postal code, user age, and baseline user depression scores. To determine whether users resided in rural or urban areas, postcodes were categorised according to the Australian Government’s postcode delivery classification system [38]. Postcodes that had both rural and urban classifications were designated as urban. Records with missing or incorrectly entered postcode data were excluded from the analysis.

For *MindMum*, *What Were We Thinking*, and *BabySteps*, we collected data on total website visits and total registrations, respectively.

### 2.6. Promotion to Consumers and Stakeholder Organisations

The project employs a multi-faceted promotional strategy to increase the visibility and uptake of *MumSpace* and its associated resources, including *MumMoodBooster*, among both professional and public audiences. This strategy targets consumers within the perinatal population as well as health professionals and relevant organisations through various communication channels.

For consumer outreach, a range of digital engagement pathways were utilised, including social media platforms such as Instagram and Facebook. Data on consumer interaction were extracted from these platforms, tracking metrics such as views, impressions, likes, clicks, comments, follows, and shares. These insights informed the iterative refinement of targeted messaging and audience engagement strategies, ensuring a responsive and dynamic promotional approach. Facebook advertising and Instagram stories were also utilised to reach younger audiences, with specific campaigns promoting mental health support and early intervention.

To extend outreach to health professionals and stakeholder organisations, targeted engagement strategies were developed, which led to a number of direct collaborations and partnerships (see Results). Key health professionals targeted included the E-Mental Health in Practice (eMHPrac) team, who undertook promotion to practitioners through training, listing on their website (www.emhprac.org.au: accessed on 19 February 2025), and inclusion in their resource handbook, Maternal and Child Health Services, perinatal health agencies, General Practitioners (GPs), consumer advocacy organisations, mental health promotion agencies, and Primary Health Networks (PHNs: in Australia, around 30,000 GPs are grouped regionally into 31 PHNs for administrative and service delivery purposes). In addition to digital outreach, community engagement was fostered through participation in expos and other public events, where we distributed posters, postcards, and other promotional materials. This expanded the reach to offline audiences. Mail-outs targeted key consumer groups, and media releases were disseminated to increase visibility across traditional and digital news platforms.

Our promotional efforts to consumers included a comprehensive social media campaign, with regular posts on Facebook and Instagram, as well as the use of paid advertising to increase reach and engagement. Posts were strategically scheduled to align with key perinatal health awareness dates and tailored to target expectant and new parents. We posted regular content every 2–3 days, maintaining continuous visibility through ongoing Facebook and Google advertisements, which have attracted widespread engagement and sharing.

The promotion efforts also included advocacy activities, with submissions to state and federal government departments to explore potential expansions of the programmes offered through MumSpace.

Metrics from each promotional channel were tracked to assess reach and engagement, helping guide future strategies and ongoing refinement of communication pathways.

### 2.7. Data Visualisation and Analysis

In preparing the data from *MumMoodBooster*, *Mum2BMoodBooster*, *MindMum*, and *BabySteps*, we conducted thorough data cleaning, which included the removal of any test accounts, duplicate entries, and incomplete records. Data are presented in tables and in graphical form as line graphs or bar charts. Where appropriate, mean values are reported with accompanying ranges or standard deviations.

## 3. Results

### 3.1. MumSpace Website

We have Google Analytics data available for MumSpace from 2017 to 2024. However, due to the introduction of Google Analytics 4 (GA4) in 2022, not all Google metrics line up across the two versions of Google Analytics. Consequently, we could only extract the most up-to-date sets of variables for the period between 2022 and 2024. Reporting periods are indicated for each set of variables below.

#### 3.1.1. Visits

From launch in late 2017 through to June 2024, the *MumSpace* site had 218,678 sessions (visits) in total. *MumSpace* has continued to successfully channel users to our stepped-care resources, with the number of visitors to the website each year exceeding that of the preceding year (Figure 3).

#### 3.1.2. Usage Statistics

Below are the usage statistics for *MumSpace* including, but not limited to, page views from 2017 to 2024 (Table 1), gender, number of resource downloads, and device type between 2022 and 2024 (Table 2).

Of note, 22–24% of sessional visits each year were by males, and the most common devices used to access the website were mobile smartphones. The average session durations in both years (2 min 4 s and 1 min 50 s, respectively) were higher than those for the average website [39].

As shown in Table 3, apart from the home page, the most frequently visited pages on the website were ‘Online Treatments’, housing links to the *MumMoodBooster* treatment programmes, and the ‘Self-Assessment’ page containing information on understanding perinatal depression. The high visitation rate to ‘Online Treatments’ may indicate that, as intended, the website successfully directs depressed women in the target population to the *MumMoodBooster* treatment programmes. Indeed, between 2023 and 2024, there were 4161 outbound clicks to *MumMoodBooster* and *Mum2BMoodBooster* from *MumSpace*, by far the highest outbound clickout rate of any website page.

### 3.2. Visitor Acquisition

Table 4 shows the pattern of session acquisition across the study period.

The most common routes by which sessions were initiated on the *MumSpace* website were via organic searches, direct traffic to the *MumSpace* URL, social media, and referrals via links on external websites (Table 4). The data on direct traffic suggest that our wider marketing had familiarised visitors with the *MumSpace* name and URL, which is consistent with the observed increase in both direct traffic and acquisition via social media across this period. The most common source of acquisition from external websites (non-social media) was the Raising Children website (https://raisingchildren.net.au/: accessed on 19 February 2025, 2180 in total across this period).

### 3.3. Geographic Locations of Usage Across Australia

Table 5 shows that sessions on the *MumSpace* website reflect the geographic distribution of Australia’s population. In rank order of size, the respective populations of Australia’s states and territories are the same as listed in Table 5. As percentages of the national population, the figures in Table 5 are also closely commensurate with the geographic distribution of the country’s population. This strongly suggests that the reach and usage of *MumSpace* is nationally consistent in proportional terms.

### 3.4. Stepped-Care Resources on MumSpace

#### 3.4.1. BabySteps

Between 2017 and June 2024, there have been 2479 registrations with *BabySteps*, with users having an average age of 33.79 (SD = 7.85) years. Users discovered *BabySteps* through various channels, including parenting websites (32.31%), web search (31.10%), hospitals (13.07%), medical practitioners (10.89%), friends or family members (10.37%), and magazines (2.26%).

Between 2022 and 2024, 88% of registrants were female, and 12% were male.

#### 3.4.2. What Were We Thinking

Users do not formally register for using the WWWT website. We therefore report visits to the site (number of times the website was opened). Between October 2017 and June 2024, there were 67,203 visits from Australia to the website. As with *MumSpace*, most of these visits were from Victoria and New South Wales.

There were also 46,233 visits from other countries (WWWT is promoted internationally).

#### 3.4.3. MindMum Smartphone App

From 2017 to June 2024, *MindMum* saw 9711 downloads via the Apple App Store (iOS) and 4462 downloads via the Google Play Store (Android).

Since July 2020, *MindMum* has been able to record registrations. From that time to June 2024, there were 5764 registrations, with 35% on Android and 65% on iOS devices. Further data on usage of the app will be published separately.

### 3.5. Digital Treatment Programmes

*MumMoodBooster* and *Mum2BMoodBooster* have had 8145 and 2357 registrations, respectively, from late 2017 to June 2024. Of the *MumMoodBooster* registrations, 7184 were women accessing the programme for themselves, 100 were health professionals, 451 were other, and 410 were unknown. Of the *Mum2BMoodBooster* registrations, 2158 were women accessing the programme for themselves, 36 were health professionals, 77 were other, and 86 were unknown.

On average, postnatal women who registered for *MumMoodBooster* were 32.08 years old (SD = 4.68) and had an average EPDS score of 16.47 (SD = 4.62). The average age of the women’s babies can be seen in Table 6, with the majority falling between less than 1 month and 24 months of age.

In comparison, women using the antenatal *Mum2BMoodBooster* programme were typically 31.60 years old (SD = 4.90) and 21.95 weeks pregnant (SD = 9.20), with an average EPDS score of 16.97 (SD = 4.96). Improvements in depression scores are similar to results from previous randomised controlled trials of *MumMoodBooster* [24,25]. The average age of registrants in both programmes is very close to the national average of women giving birth in Australia (31.2 years; [40]). The average EPDS scores of registrants in both programmes are substantially above the cut off score (≥13) for detecting probable depression recommended in Australia [41]. These statistics suggest that *MumMoodBooster* and *Mum2BMoodBooster* are both reaching depressed women in the target demographic. Using postcodes, *MumMoodBooster* and *Mum2BMoodBooster* users were categorised as either living in rural or urban Australia [38]. Data were available for 7986 *MumMoodBooster* registered users. Of these, approximately 32% reported residing in rural or remote postcodes, while 68% were classified as residing in urban areas across Australia (see Figure 4). Similarly, for *Mum2BMoodBooster*, 2262 users reported their postcode, 29% reported residing in rural or remote postcodes, while 71% were classified as residing in urban areas across Australia (See Figure 5).

### 3.6. Promotion to Consumers and Stakeholder Organisations

#### 3.6.1. Consumers

Our promotion strategy was accompanied by increased engagement across consumer audiences.

##### Facebook

From December 2018 to 30 June 2024, the *MumSpace* Facebook page had 6229 followers. Between 2023 and 2024, the reach of organic or paid distribution of *MumSpace* Facebook content, including posts, stories, and ads was 407, 982, with 10,562 link clicks. See Figure 6 below for a breakdown of the age and gender of these followers.

##### Instagram

As of 30 June 2024, the *MumSpace* Instagram page had 1547 followers. See Figure 7 below for a breakdown of the age and gender of these followers.

#### 3.6.2. Stakeholders

##### Early National Exposure

The initial launch in 2017 was endorsed and announced by the Federal Minister of Mental Health, a strategic action which generated high-level visibility and credibility across published media. Subsequently, *MumSpace* won the Victorian Premier’s Award for Improving Maternal, Child, and Family Health in November 2019. This award further generated media coverage and interest on radio and publishing and digital platforms.

##### Uptake by Primary Healthcare Providers

*MumMoodBooster* has been successfully incorporated into several Primary Health Network (PHN) HealthPathways platforms (HealthPathways is a digital platform used by most PHNs in Australia, and also in New Zealand and the UK, to assist GPs in deciding on best-practice care options and in locating locally accessible services: https://www.healthpathwayscommunity.org; accessed on 19 February 2025). Our efforts generated collaborations with North-Western Melbourne and Murray (regional Victoria) PHNs to achieve several key outcomes. First, we promoted *MumSpace* and its resources, focussing on the adoption of *MumMoodBooster* and *Mum2BMoodBooster* as free, evidence-based digital treatments for postnatal and antenatal depression, respectively. Second, we engaged general practitioners in these PHNs for pilot testing of a new clinician portal. Lastly, we enhanced current perinatal mental health HealthPathways platforms, integrating *MumMoodBooster* and *Mum2BMoodBooster* as key treatment options within these frameworks.

In addition, we strengthened relationships with over 10 PHNs, including Gold Coast, Central and Eastern Sydney, and North-Western Melbourne. Notably, Central and Eastern Sydney PHN has added *MumMoodBooster* to their new mental health service directory, enabling GPs and other healthcare professionals to access these programmes quickly and effectively. We were published in over 10 PHN newsletters, with North-Western Melbourne PHN further promoting the *Mum2BMoodBooster* Clinician Portal through their GP networks. Our ongoing promotion through professional health networks has significantly increased referrals to *MumSpace*, resulting in over 26,000 website visits in 2022 and 2023 alone.

##### Relationships Built with Relevant Agencies and Providers

Beyond PHNs, we have also targeted health professionals via other routes. General practitioners were engaged through webinars, while maternal and child health nurses were reached via conference presentations and newsletters. For instance, we exhibited at the Victorian Maternal and Child Health Conference in 2023, sharing promotional materials with attendees and publishing articles in the Victorian MCHN newsletter. Psychologists also participated in a pilot study involving *MumMoodBooster*, providing valuable feedback on effectively integrating the programme into psychological practice.

Our community engagement has been further strengthened through health promotion efforts supported by tailored educational resources for health professionals, including digital newsletters and conference presentations. A key component of this strategy was the creation of downloadable resources, such as the *MumMoodBooster* Clinician Portal brochure, which has been downloaded 1305 times since November 2023, and a quick referral guide for *MumMoodBooster* programmes, downloaded 659 times since May 2023. These resources have fostered new collaborations, including discussions with obstetricians and key health bodies such as WA Health Promotions Network and Queensland Health, helping to integrate *MumSpace* with existing healthcare services across Australia.

Promotion to these groups involved a mix of digital and traditional media approaches. Targeted social media campaigns via LinkedIn, Facebook, and Instagram provided professional audiences with up-to-date information on *MumSpace* and its resources. Our LinkedIn campaign alone, launched in November 2023, achieved an organic reach of 37,805, with 1411 post reactions and 1323 clicks. We strategically targeted television, radio, and news segments to increase public awareness of perinatal mental health resources.

Direct marketing to health agencies involved the use of digital newsletters and tailored content to ensure the information reached relevant professional networks. Specialised webinars and presentations at national and state conferences provided platforms for professional engagement, enabling deeper discussions of *MumSpace*’s impact and efficacy. Distribution of hard copy and digital information packages to health agencies reinforced the availability of resources for perinatal mental health support. For instance, in April, Gold Coast PHN ran a Maternity Alignment Program that distributed copies of our clinician portal brochures, setting a precedent for future partnerships.

Direct approaches to key leaders in health and advocacy organisations facilitated collaborations and the broader promotion of the programme’s tools within existing health infrastructure. Collaborative partnerships with organisations like the Gidget Foundation, SMS4dads, and ForWhen have strengthened referral pathways, enhancing service connectivity.

The impact of our promotional strategies, combined with ongoing collaboration with PHNs, health professionals, and the community, has been overwhelmingly positive. These efforts have been accompanied by substantial increases in traffic to the *MumSpace* website, with over 15,000 visits from partner websites in 2023 alone, and greater utilisation of the available resources. *MumSpace* and its resources including our treatment programmes have recently been included on the Centre of Perinatal Excellence (COPE) e-Directory.

### 3.7. New Developments in the Study Period: Implementation and Maintenance

Key to the *MumSpace* initiative is the implementation of its resources, particularly our core treatment programmes, into practice.

#### 3.7.1. Clinician Portal

The recently developed *MumMoodBooster* and *Mum2BMoodBooster* Clinician Portal serves as an example of our quality improvement activities and is an integration and implementation into clinical practice, driven initially by engagement with General Practitioners and now including multidisciplinary clinicians, and a response to low referral rates. The clinician portal has been developed in order to promote and encourage health professionals to incorporate *MumMoodBooster* as part of referral pathways in the treatment and management of perinatal anxiety and depression in the wider community. It can be accessed via a landing page developed within the *MumSpace* website for health professionals, directly linking them to the registration page for the respective portal. The portal was developed with extensive user testing and consultation with clinicians and is being actively promoted across selected Primary Health Networks. The clinician portal for *Mum2BMoodBooster*, our treatment programme for antenatal women, was subsequently developed.

Since the *MumMoodBooster* Clinician Portal was rolled out in 2020, there have been a total of 349 clinicians who have registered for an account. The *Mum2BMoodBooster* Clinician Portal was officially rolled out in 2023 and has seen 49 clinicians register.

See Table 7 below for a breakdown of the types of clinicians that have registered.

#### 3.7.2. Quality Improvement Activities

Based on stakeholder feedback, continual modifications are made to *MumSpace*. For example, additional *MumSpace* resources were developed (consumer factsheet, new tip sheets, a poster to promote *MumSpace* in child and family health services, and an e-card) available for download via the *MumSpace* website. The introduction of a new ‘Resources’ tab provides access to these new downloadable and printable resources, for both health professionals and consumers.

In addition, given the changing technology and in response to user feedback, we will soon release a new version of *MumMoodBooster* with increased functionality and responsivity on mobile phones. A similar process is underway for *MindMum*.

The improvement in *MumMoodBooster’s* responsivity on smartphone devices and search engine optimisation, user-flow, and the onboarding experience of *MumSpace* form part of the ongoing maintenance of the initiative and its key resources.

## 4. Discussion

Between 10 and 20% of women will experience depressive symptoms in pregnancy or in the first postnatal year [18]. In Australia, approximately 300,000 women give birth every year [40], which equates to between 30,000 and 60,000 new cases annually. In 2017, we set out to deliver, and maintain, evidence-based resources and digital treatments for these women in a nationally accessible, stepped-care model that services various levels of symptomatology, free at the point of use. This is seen as a sustainable approach to the delivery of such services (Mendes-Santos 2022; National Mental Health Commission 2016). As of 2023, 96% of the Australian population could access the Internet [42], and smartphone penetration in Australia was estimated to be 86% [43], indicating that the basis for access to digital mental health support is broad.

Data since 2017 indicate that we have successfully addressed our key objective of providing early and readily accessible support for a substantial fraction of women with, or at risk of, perinatal depression in Australia. The *MumSpace* website has been accessed by over 275,000 visitors, and approximately 10,000 depressed women have engaged with our digital *MumMoodBooster* depression treatment programmes. Many others have benefitted from our universal and preventative resources. These represent a substantial fraction of the target population.

Internationally, other efforts to translate and maintain effective national digital services of this kind have had mixed results and have encountered both successes and encountered various barriers, which we also experienced (for example, [44]). As concluded by Mohr and colleagues [45], “*There is an enormous research-to-practice gap in digital mental health, with strong and growing evidence from efficacy trials … yet virtually no successful and sustainable implementation.*” Our own experience concurs with the research-to-practice gap. Nevertheless, we employed strategies that allowed us to sustain national delivery of our services for close to a decade. Whilst ongoing development to overcome emerging challenges remains necessary, our efforts to date reflect the feasibility of successful promotion and widespread uptake among our target audience.

Our targeted marketing campaigns, including social media and stakeholder engagement strategies aimed at helping health professionals to effectively use the *MumSpace* tools in practice, appeared to significantly boost visibility and engagement nationally. We recognise that, clearly, not every woman in the target population is accessing MumSpace, and ways of encouraging women to access help and overcome barriers to engagement remains a major challenge in this field. At the centre of our model are the *MumMoodBooster* and *Mum2BMoodBooster* digital CBT treatments. Around these, we provide various types of stepped support and programmes of varying intensity designed to service differing levels of need including universal and prevention programmes. These early self-help interventions aim to reduce and ameliorate the impact of perinatal depression and to reduce the burden on more traditional acute downstream services.

The “Digital Treatments” page of the central *MumSpace* website was found to be the most visited page. The average age of the users of our digital CBT programmes is closely commensurate with the age of women giving birth in Australia [40], indicating that the target demographic is being reached across the nation. In addition, the mean postnatal EPDS score was 16.47 (SD = 4.62), showing that the majority of registrants were close to or well above the threshold of ≥13, commonly considered as indicating probable depression [46]. In addition to women, our analytics tell us that the website was visited by 22% of consumers who identify as male. National reach was broadly proportionate to the distribution of Australia’s population between states and between urban versus rural communities. Our figures showed that most users used a mobile device.

We were also able to develop strong relationships and collaborations with health professionals and agencies and begin initiatives to integrate new innovations into practice. Close to one-third of Australia’s Primary Health Networks now list the *MumSpace* resources on their digital assets. However, ongoing monitoring and quality improvement to increase reach and engagement are a constant requirement.

## 5. Strengths and Limitations

The study reported here is a real-time service evaluation. As such, there is no control condition against which to judge our results. Data sources such as Google Analytics are limited and, to some extent, somewhat opaque as to the identity and characteristics of users. Challenges included changing technology and the fact that those responsible for developing services and those responsible for implementing technological delivery solutions often come from disparate professions and backgrounds, which may result in workflow communication obstacles (as found by [47]). Another challenge is the need to be responsive to any outage problems that arise, responding to user queries and monitoring of all functionalities. We addressed these challenges through weekly or fortnightly meetings, working closely on problem solving at every step. In addition, we built an interdisciplinary team with the Perinatal Depression e-Consortium at its core, including close relationships with our technology specialists as changes are now taking place at such a rapid pace that there is a real and ongoing issue of maintaining digital interventions so that they do not come rely on obsolete and incompatible technologies [45]. Other challenges included the difficulty of recruiting professionals to directly engage mothers in using the *MumSpace* resources. The majority of visits to the main website were through women directly. Steps taken to address these challenges included motivational methods to increase engagement and the development of a portal for clinicians.

As concluded by others [48], we have learned that the sustainability of population-level digital mental health initiatives requires systematic promotion and engagement strategies for success, maintenance, and scalability, as well as substantial funding and a multidisciplinary team.

## 6. Conclusions

*MumSpace* represents the delivery and implementation of a digital stepped-care approach to reduce and ameliorate the impact of perinatal depression and anxiety in Australia, bringing together a range of evidence-based tools, resources, and treatment programmes.

Our experience of delivering this national digital mental health initiative reveals a number of findings that have significance for the successful delivery of such services here and in other countries. First, with some planning and successful promotion to consumers and stakeholders, national delivery can be achieved with equitable geographic reach. Secondly, with vigorous and targeted promotion, the services can be delivered to a sizeable fraction of the depressed or ‘at-risk’ perinatal population to a relatively precise degree in terms of age, stage of parenthood, and symptom severity. Thirdly, sustained delivery of digital perinatal mental health interventions requires substantial resources in terms of finance, staffing, maintenance of and updates to technology, and a systematic approach to promotional campaigns increasing visibility to relevant audiences. Crucially, the building and ongoing maintenance of collaborative relationships with other agencies in the relevant field is a time-consuming but necessary activity for sustained success.

Our work also points to the need for further development and refinement. For example, economic evaluation of how best to assess the cost–benefit profiles of digital mental health solutions is not advanced, particularly for perinatal populations as the impact includes costs relating to partners and infants [11]. What is the return on such investments of public capital? The issue of affordability to consumers at the point of access is one very important consideration but, since health budgets are finite, the cost effectiveness for funders and their ability or willingness to pay must also be considered. We are currently working with others to develop such models [11].

Future work could usefully focus on a number of other important areas. Understandably, most previous work in digital perinatal mental health has focussed on delivering support for mothers. However, men are also vulnerable to perinatal mental health issues [49], and there are real human and economic costs associated with this [50]. Specific digital resources and treatment programmes aimed at this group could therefore be of real benefit on a population level (indicated by our own finding that men do indeed access the MumSpace website).

Furthermore, in treating and supporting families affected by symptoms of perinatal mental illness, there is a very strong case for focussing on the lifetime impact on infants [11].

Next, in diverse multicultural communities and countries, not all interventions are appropriate, or desirable, for all populations. Much work remains to be conducted in terms of increasing access, acceptability, and trustworthiness for e-mental health solutions offered to indigenous, migrant, and LGBTQI+ communities, for example.

Finally, our experience of implementation and adoption by health services suggests that more work can profitably be undertaken to build on interactive access for clinicians through our portals, being capable of monitoring patient progress and delivering “blended” models of care which hybridise digital and traditional face-to-face services [51].

We have profiled our resources on professional platforms such as HealthPathways and engaged in numerous promotional events as described. However, the relatively poor adoption rates of our clinician portal by clinicians suggest that for more comprehensive population-wide rollout, which reaches vulnerable and underserved populations, systemic changes are required. Integration with health systems would require addressing clinician barriers, organisational barriers, and influencing policy. There is huge potential to reach a much larger population given that almost all women giving birth are in contact with health services during this time.

Our intention with this current paper has been to provide an overall report of the development, reach, and implementation of the digital portal MumSpace.com.au. We will continue to evaluate the impact of the *MumSpace* resources and the ways in which they are influencing the mental health of women who are pregnant and caring for infants in Australia.

We intend to report more details in future papers dealing with the reach and clinical impact [52] of the *MumMoodBooster* depression treatment programmes and the development and uptake of the *MindMum* mobile app.

In conclusion, whilst not all perinatal consumers prefer to access their mental health support digitally, many others do not access any treatment. Our findings strongly suggest that digital delivery can effectively reach many of those in need with evidence-based resources and treatments.

Our model may inform the development and delivery of similar services so that a robust evidence base can be built to inform policy and implementation for digital perinatal mental health treatment.

## Figures and Tables

**Figure 1 ijerph-22-00361-f001:**
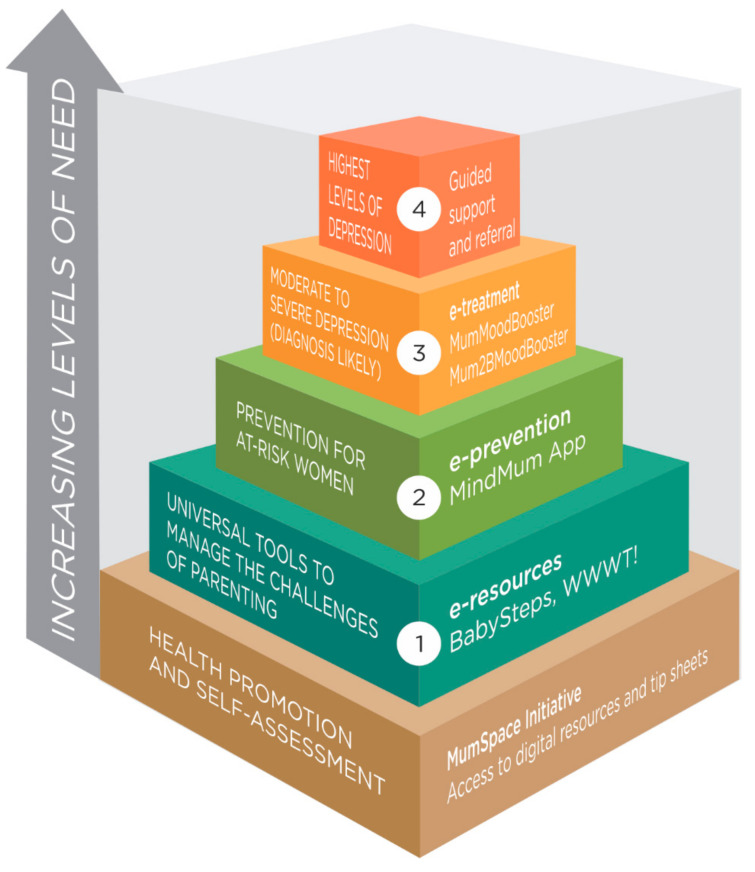
Model of care.

**Figure 2 ijerph-22-00361-f002:**
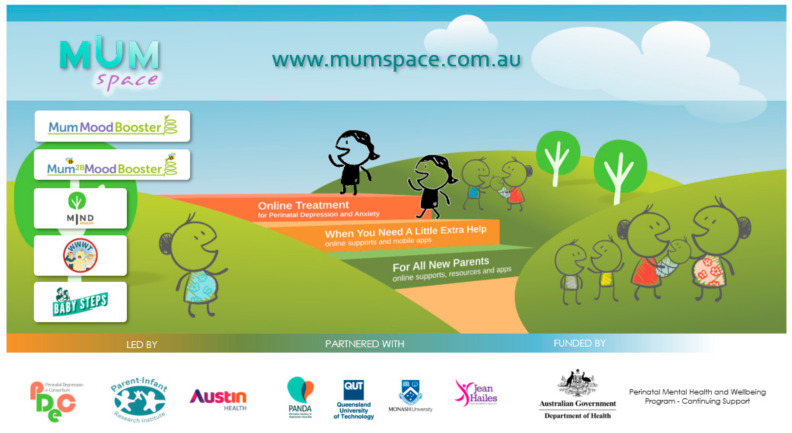
Website interface and structure.

**Figure 3 ijerph-22-00361-f003:**
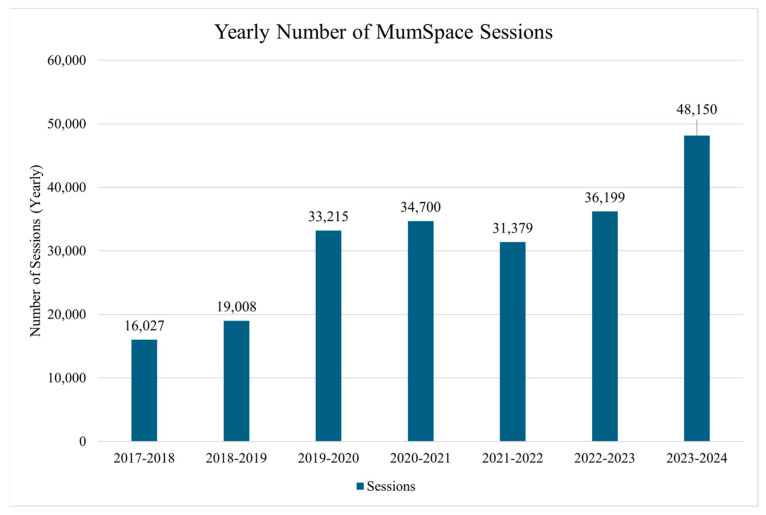
Visits to *MumSpace* since launch. Year defined as 1 July to 30 June of the following year, with the exception of 2017–2018, which is from October 2017 (launch of *MumSpace*) to 30 June of 2018.

**Figure 4 ijerph-22-00361-f004:**
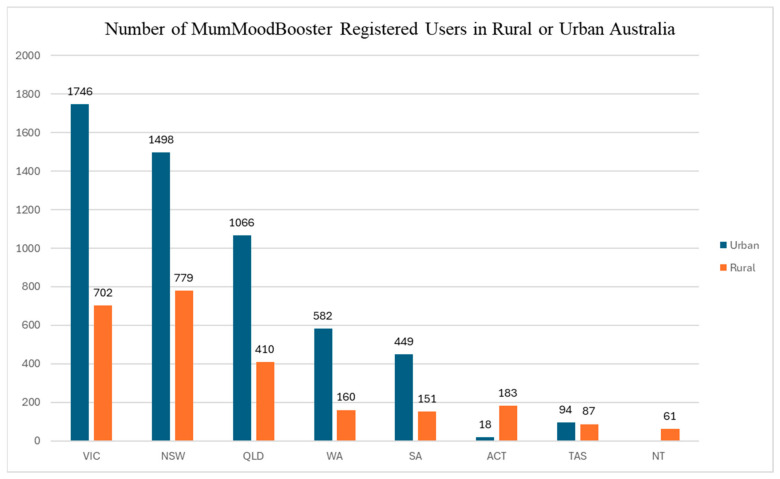
Number of *MumMoodBooster* registrants in rural or urban areas across Australia.

**Figure 5 ijerph-22-00361-f005:**
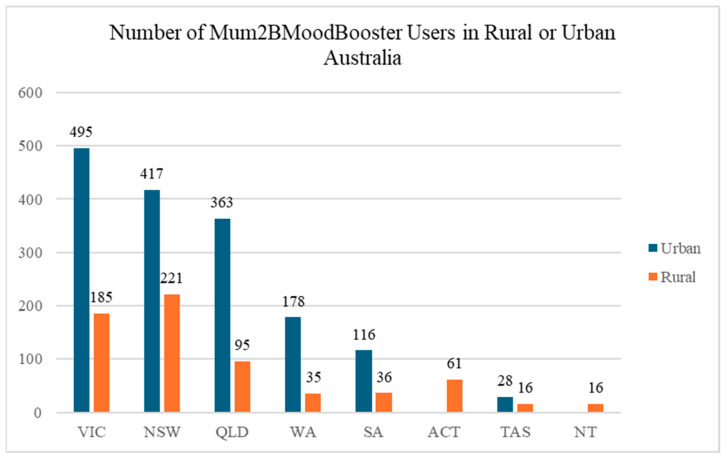
Number of *Mum2BMoodBooster* registrants in rural or urban areas across Australia.

**Figure 6 ijerph-22-00361-f006:**
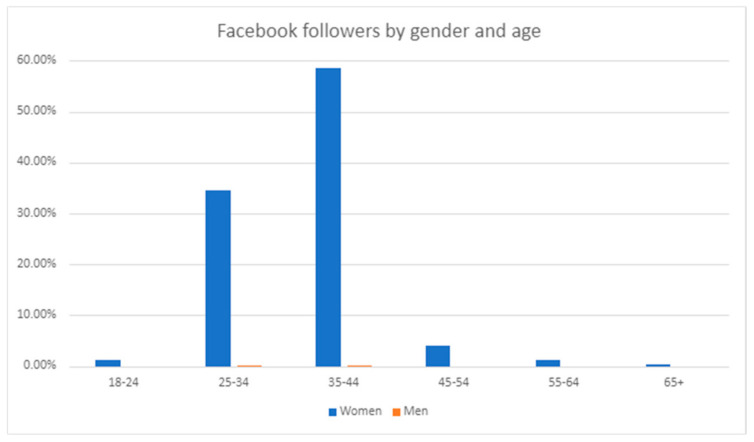
Age and gender of *MumSpace* Facebook followers.

**Figure 7 ijerph-22-00361-f007:**
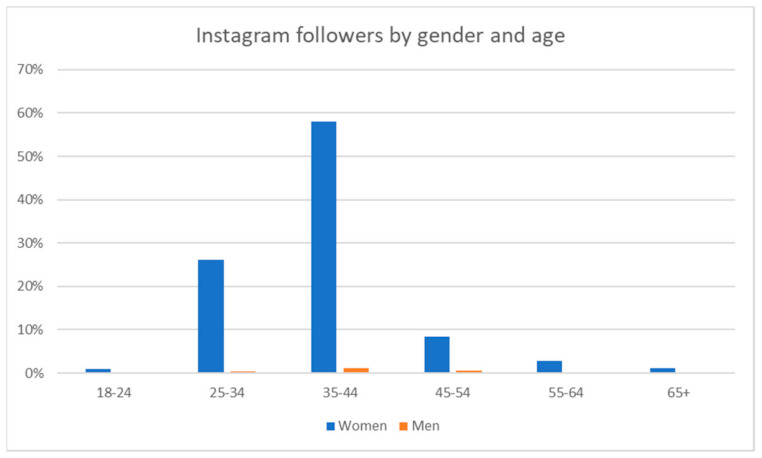
Age and gender of *MumSpace* Instagram followers.

**Table 1 ijerph-22-00361-t001:** Number of page views and average pages viewed per session between October 2017 and June 2024.

Year ^a^	Page Views	Avg Pages Viewed Per Session
2017–2018	36,149	2.46
2018–2019	39,327	2.08
2019–2020	64,655	1.95
2020–2021	59,851	1.70
2021–2022	58,309	1.80
2022–2023	40,769	1.19
2023–2024	78,661	1.65

^a^ Year defined as 1 July to 30 June of the following year, apart from 2017–2018, which is from October 2017 (launch of MumSpace) to 30 June of 2024.

**Table 2 ijerph-22-00361-t002:** Usage statistics for *MumSpace* website between 1 July 2022 and 30 June 2024.

Year	Sex (%)	Number of Downloads from Resources Page	Device Type Used to Access *MumSpace*
2022–2023	Female 78% Male 22%	2971	Smartphone: 67% Desktop/Laptop: 33% Other: <1% Unknown: <1%
2023–2024	Female 75% Male 24	3213	Smartphone: 57% Desktop/Laptop: 42% Other: <1% Unknown: <1%

**Table 3 ijerph-22-00361-t003:** Most visited pages on the *MumSpace* website based on page views (1 July 2022 to 30 June 2024).

Page	Views (n)
Home	84,543
Online Treatments	17,849
Self-Assessment	11,077
When You Need Extra Help (MindMum)	7011
For All New Parents (BabySteps and What Were We Thinking)	6435
About MumSpace	5540
MumSpace Resources	5016
Clinician Portal	2196
Urgent Help	1663

**Table 4 ijerph-22-00361-t004:** Visitor acquisition between 2022 and 2024 for *MumSpace*.

	2022–2023 *	2023–2024 *
Organic search	7856	9576
Direct traffic	7471	10,824
Social media	5727	7387
Referral from listings on external websites	3785	3867
Other	203	305
Google Ads (grant programme) starting from 16 August 2023		8132
Paid Facebook adverts	6156	8059

* Year defined as 1 July to 30 June of the following year.

**Table 5 ijerph-22-00361-t005:** Percentage of total sessions by geographic location.

Region	Percentage of Total Sessions (%)
Australian Regions	
New South Wales	29
Victoria	25
Queensland	19
Western Australia	9
South Australia	6
Tasmania	2
Australian Capital Territory	1
Northern Territory	<1
Regions outside Australia	7
Unknown	2

**Table 6 ijerph-22-00361-t006:** Frequency and mean (SD) of babies’ ages in months for women registered with *MumMoodBooster*.

	Frequency of Baby’s Age (in Months) by Category	Mean (SD) of Baby’s Age (in Months)
<1 month old	927	N/A
1–24 months old	5939	6.02 (4.9)
25–48 months old	194	32.07 (6.4)
>48 months old	54	N/A

**Table 7 ijerph-22-00361-t007:** Clinician registrations to portal.

Profession	MMB Clinicians (n)	Mum2B Clinicians (n)
Nurse	130	11
General Practitioner	87	11
Psychologist	60	9
Social Worker	21	7
Midwife	18	2
Occupational Therapist	7	1
Counsellor	5	0
Researcher	3	0
Obstetrician	1	1
Other	7	3
Total	339	45

## Data Availability

The original contributions presented in this study are included in this article. Further inquiries can be directed to the corresponding author.

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
