# Peer review of "National Implementation of Perinatal Mental Health Treatment—The MumSpace Digital Stepped-Care Model"

_ijerph, 2025, doi:10.3390/ijerph22030361_

Round 1
Reviewer 1 Report
Comments and Suggestions for Authors
Dear Authors.
I am grateful for the opportunity to review your work. After reviewing and analysing your work, I would like to share some thoughts with you. The description and scope of the MumSpace initiative has been very comprehensive, not only in reference to the tool and the resources it hosts but also the usefulness given to it by the participating users. Congratulations on the work.
- In terms of the population's response, it is noted that it was very well received. However, postpartum and even prepartum depression is a topic that is often kept secret and difficult to detect. Could this be a limitation? Although MumSpace has had a significant reach among the population, could there be a part of the population that has not actively participated in the tool?
- The fact that access figures have multiplied in recent years is a reflect statistic, coupled with the fact that the second most visited page on MumSpace is the online treatment page. In your opinion, do you think this is because more and more women are involved in this situation and are asking for help?
- The results show that by far the oldest age group of MumSpace followers on social media is around 35-44, followed by the previous stage. Could this be because the age of motherhood has recently been in that range? Perhaps they are also the biggest consumers of social media content.
- Finally, I find it particularly interesting the percentage of men who visit the site, about one in 4-5. In the conclusions of the paper they mention it slightly but I think it would be very interesting to describe what perinatal health problems they suffer from or perhaps if the fact that their partner may be going through perinatal depression has an impact on them as well.
Best regards
Reviewer 2 Report
Comments and Suggestions for Authors
The authors have presented an important study about "National Implementation of Perinatal Mental Health Treatment – ​​the MumSpace Digital Stepped-Care Model".
In order to improve the quality of the manuscript, it is necessary for the authors to consider the following corrections and suggestions for further information:
1. The study presented does not include a characterization of the population that uses the MumSpace digital stepped-care model. This information is necessary to determine usability factors related to age, educational level, digital divide, among others.
2. It is necessary to compare the results related to the use of the app and the use of the WEB. If there is a correlation, it should be explained in the results section.
3. The authors should propose a profile of the ideal professional who should care for mothers. Each profile should vary according to the variables of the characterization of the population.
4. Figures 6 and 7 are very blurry.
5. Figure 2 is not a web architecture. It should be an interface
6. Update several references considered obsolete:
ref 1 from 2005
ref 7 from 2004
ref 28 from 1996
ref 31 from 1999
ref 33 from 2001
ref 56 from 2006
